# Fruit and Vegetable Consumption, Household Food Insecurity, and SNAP Participation Among Attendees of Free Produce Events at Safety-Net Health Center Sites

**DOI:** 10.3390/nu17111849

**Published:** 2025-05-29

**Authors:** Julia I. Caldwell, Fatinah Darwish-Elsherbiny, Keisha Macon, Gloria Moon, Alejandra Casillas, Arleen F. Brown, Dipa Shah, Tony Kuo

**Affiliations:** 1Nutrition and Physical Activity Program, Division of Chronic Disease and Injury Prevention, Los Angeles County Department of Public Health, 3530 Wilshire Blvd., 8th Floor, Los Angeles, CA 90010, USA; jcaldwell@ph.lacounty.gov (J.I.C.); fdarwish@ph.lacounty.gov (F.D.-E.); kmacon@ph.lacounty.gov (K.M.);; 2Division of General Internal Medicine and Health Services Research, David Geffen School of Medicine at UCLA, 200 Medical Plaza, Suite 420, Los Angeles, CA 90095, USA; gmoon@mednet.ucla.edu (G.M.); abrown@mednet.ucla.edu (A.F.B.); 3Community Engagement and Research Program, UCLA Clinical and Translational Science Institute, 10833 Le Conte Ave., BE-144 CHS, Los Angeles, CA 90095, USA; 4Department of Family Medicine, David Geffen School of Medicine at UCLA, Box 957087, Los Angeles, CA 90095, USA; 5Department of Epidemiology, UCLA Fielding School of Public Health, Box 951772, Los Angeles, CA 90095, USA; 6Population Health Program, UCLA Clinical and Translational Science Institute, 10833 Le Conte Ave., BE-144 CHS, Los Angeles, CA 90095, USA

**Keywords:** public assistance, free produce events, fruit and vegetable consumption, household food insecurity, SNAP participation

## Abstract

Background/Objectives: Safety-net health centers are increasingly screening for food insecurity and providing patients with referrals to public assistance programs—e.g., the Supplemental Nutrition Assistance Program (SNAP). However, not all individuals actively participate in or are eligible for these programs. Onsite distributions of free produce at health center sites represent a promising complementary option for addressing this need. This study examines free produce events at these sites and their associations with attendees’ food and vegetable consumption, household food insecurity, and SNAP participation (study outcomes). Methods: In 2024, an intercept survey was conducted with 497 adults attending produce events at 16 safety-net health center sites in Los Angeles County, California, USA. Descriptive analyses profiled these food events, gathering information on attendee characteristics. Multivariable regressions examined associations between event attendance and study outcomes. Results: Over 80% of attendees lived in food-insecure households. Among those who were patients of the event site, 68% and 28%, respectively, indicated they received information about Medicaid and SNAP from the clinic staff. Compared to first-time attendees, those who attended produce events frequently consumed, on average, one additional serving of fruit and vegetables a day (*p* < 0.000). Conclusions: Offering free produce events at health center sites, where many attendees receive usual care, is a promising strategy for increasing healthy food access among safety-net populations. This underutilized approach is a viable intervention for improving access to healthy food, especially in circumstances where patients are not eligible for public assistance or nutrition incentive and/or food pharmacy programs are not readily available.

## 1. Introduction

Approximately 13.5% of households in the United States (U.S.) are food-insecure, with 48.6% of U.S. adults having some form of cardiovascular disease and 11.3% with diabetes, much of which can be attributed to poor diet [1,2,3]. Recognizing their unique role in addressing social determinants of health such as food insecurity, safety-net health systems (e.g., low-income community clinics and Federally Qualified Health Centers) are increasingly screening for food insecurity and referring eligible patients to food resources—e.g., the Supplemental Nutrition Assistance Program (SNAP), local food pantries, and onsite food distribution events.

Although screening for food insecurity has been successfully integrated in many clinical environments, the same cannot be demonstrated for referring patients to relevant food resources (e.g., federally funded food assistance programs and food banks) after patient needs have been identified. Factors such as competing time demands on clinic staff, lack of knowledge about the eligibility requirements of programs, access to appropriate wraparound services, and limited coordination of social services all impede optimal utilization of these strategies [4,5,6]. Challenges with low patient participation in traditional public assistance programs, such as SNAP, remain despite emerging research showing the positive impacts of SNAP on health outcomes, food security, and healthcare utilization [7,8].

In some U.S. safety-net health systems, alternative interventions (options) such as food pharmacies and/or onsite food distribution events (on the campuses of health center sites) have gained popularity and garnered greater acceptance as programs that can be facilitated by healthcare institutions. As “Food as Medicine” (FAM) interventions, the concept that food can be used as a primary prevention and/or complementary treatment for health conditions [9,10], these options can offer expanded access to healthy foods through the healthcare system as compared to traditional public assistance programs (including SNAP). FAM interventions range from more individualized approaches like medically tailored meals for specific medical conditions to produce prescription programs that provide vouchers for fresh fruits and vegetables; they have all been associated with improvements in patient dietary consumption, healthcare utilization, and medical spending [11,12,13]. Most patients, especially those with diet-related conditions such as prediabetes, type 2 diabetes, and/or hypertension, can access food immediately after visiting their healthcare provider, generally at the same location as the medical office, typically without a need for additional travel [14,15,16]. Patients, for instance, who visited a hospital-based food pantry expressed comfort, convenience, and less stigma, suggesting an added advantage that this onsite intervention can offer over more traditional food pantries [16]. An important aspect of these interventions is that they complement, rather than detract from, traditional public assistance; in most cases, they preserve the opportunity for eligible individuals to apply and obtain SNAP and other federally or state-funded program benefits.

Despite FAM leaders and practitioners calling for greater integration of these programs into health-related settings [17], these alternative intervention options have faced numerous challenges in their implementation. For instance, to offer affordable, quality foods, reliable partnerships and coordination with community organizations (e.g., food pantries, food banks, and others) are typically required, giving way to a substantial amount of time, energy, and funding invested [18]. Sufficient real-world data to justify program continuation to clinic leadership is another major barrier. How these interventions affect target behaviors (e.g., fruit and vegetable consumption), address household food insecurity, and assure SNAP participation (i.e., actively enrolled and using the benefits) are generally not well characterized in the literature [19,20]. This is especially so for food distribution events (i.e., facilitating free produce events at health center sites), a focus of this program evaluation.

This study addresses these gaps in nutrition and FAM practice, describing the efforts of several health centers in Los Angeles County, California, USA, to integrate onsite produce events as an intervention strategy for helping patients manage their diet-related chronic conditions. In particular, this study examines whether the frequency of attending free produce events at safety-net health center sites was associated with higher fruit and vegetable consumption, lower household food insecurity, and greater SNAP participation.

## 2. Materials and Methods

### 2.1. Setting and Context

In Los Angeles County, 41% of low-income households experienced food insecurity in the past year [21]. To help address this public health problem, the Los Angeles County Departments of Health Services (DHS) and Public Health (DPH) began screening patients for food insecurity within their ambulatory care network (i.e., safety-net health centers), hospital-based outpatient clinics, and community wellness centers—this effort started in 2019. Patients who screened positive for food insecurity were referred to relevant food resources (food pantries, food banks, SNAP, etc.) when needed. The goal of this screening and referral process was to assist the DHS and DPH in identifying patients who may benefit from dietary strategies that can improve their chronic disease management. Not too soon after the adoption of this clinic workflow, the County of Los Angeles Board of Supervisors issued a directive (Board motion) directing the DHS and DPH to continue this effort and to look for future opportunities in which patient and community access to healthy food can be expanded [22]. In response to the motion, the DHS and DPH formed the Los Angeles County Food Rx Collaborative (the “Collaborative”). This learning collaborative, launched in 2021, is designed to provide a regular forum for peer-to-peer learning and a protected space for developing and testing program strategies in the field.

In 2023, the University of California, Los Angeles (UCLA) partnered with the DHS to launch a program implementation study through the National Institutes of Health/National Heart, Lung and Blood Institute’s Disparities Elimination through Coordinated Interventions to Prevent and Control Heart and Lung Disease Risk (DECIPHeR) Alliance. The study, called the LAC HTN Equity Initiative, sought to advance equity in hypertension care across Los Angeles County, paying particular attention to the role of a healthy diet in controlling high blood pressure. To expand this community strategy, UCLA-DECIPHeR joined the Food Rx Collaborative to engage community agencies, providing them with access to research expertise and program evaluation resources. These technical assistance and resource services helped many of these agencies to further scale and improve culturally appropriate, evidence-based programming for hypertension in high need communities. Among the salient questions that the UCLA-DECIPHeR team helped answer for the Collaborative was whether or not free produce events at safety-net health center sites could lead to increased fruit and vegetable consumption, reduced household food insecurity, and/or active SNAP participation among the attendees. Between October 2020 and March 2025, 16 DHS and DPH health center sites facilitated a total of 944 onsite produce events, distributing more than 4 million pounds of produce to 831,903 people (217,791 households).

### 2.2. Data Collection

In 2024, a cross-sectional intercept survey was administered to attendees of produce events facilitated by the 16 DHS and DPH sites. Attendees were approached by evaluation staff who were at the health centers on the day of the survey; they were waiting in line for produce. Due to the large size of the events, attendees at both the front of the line and the back of the line were approached. As the events were open to the public, attendees included both community members and patients referred to these events by their healthcare providers. To satisfy eligibility, staff confirmed on paper that attendees were over the age of 18, resided in Los Angeles County, and had not completed the survey previously. Prior to starting the questionnaire, attendees were informed individually that the survey was voluntary and that participation would not affect their ability to receive free produce or other services at their usual care clinic site. The survey questionnaire was developed by the evaluation staff using adapted question items as well as some new items developed specifically for this survey. It was available in English and Spanish and self-administered using pen and paper. The evaluation staff were also available onsite to assist attendees with the survey if they ran into difficulties. All attendees who completed the survey were given a USD 15 produce gift card as a token of appreciation and to help offset costs and time spent on completing the questionnaire.

### 2.3. Outcomes

This study focused on three outcomes (dependent variables) used in its multivariable regression analyses (models). The first outcome was fruit and vegetable consumption—this outcome was assessed by combining responses from two questions in the survey: (i) “On an average day, about how many servings of fruit do you eat? (1 serving is about the size of your fist)”; (ii) “On an average day, about how many servings of vegetables do you eat?” Response categories included “None”, “Less than 1”, “1”, “2”, “3”, or “4 or more”. For analytic purposes, these responses were combined to generate a range of servings consumed, from 0 to 8. The second outcome, household food insecurity, was assessed using the six-item U.S. Food Security Module; psychometric properties and the utility of this tool are described elsewhere [23]. Attendees who completed the survey were categorized as “food secure” (scored 0 or 1 on the module) or “food insecure” (scored 2–6 on the module). Finally, SNAP participation was defined as being “enrolled and actively engaged” in this food stamps program, receiving benefits from it. In the survey, this outcome was asked as a “yes” or “no” question.

### 2.4. Independent Variables

The primary independent variable for this study was the frequency of attending free produce events at the 16 DHS and DPH health center sites. Attendees were asked “In the last 12 months, how many times have you (or anyone in your household) ever come to produce distributions at this location?” They were coded as “first time”, “occasional (2–9 times)”, or “frequent (≥10 times)” attendees.

To better understand the health center referral process, attendees who self-identified as a patient and had completed the survey were also asked about their participation in public assistance programs—i.e., whether or not they had received a referral to enroll in any of the listed programs: SNAP/CalFresh/Food Stamps, Medi-Cal, Special Supplemental Nutrition Program for Women, Infants, and Children (WIC), California Food Assistance Program, CalWorks (Temporary Assistance for Needy Families), Housing Assistance Programs (Section 8), Social Security Disability Income (SSDI), Supplemental Security Income (SSI), and/or none of the above.

### 2.5. Covariates

Study covariates included gender (female or male), age (18–34, 35–64, or ≥65), race/ethnicity (Hispanic or Latino/a/x, Black or African American, White, and Other or Multi-Race), household size (1–2 people, 3–4 people, 5–6 people, and 7+ people), SNAP participation—enrolled and participating (yes or no), and whether attendees were patients of the health center site (yes or no). Event attendees were also asked if they or anyone in their household have been diagnosed with any of the following health conditions: heart disease, prediabetes, diabetes, hypertension (high blood pressure), or overweight/obese. Attendees who responded affirmatively, indicating they have at least one chronic health condition, were coded as having a “household chronic disease”, whereas attendees indicating they had none were coded as not having a “household chronic disease”. Because this variable was asked at the household level, it was not included in the multivariable regression analyses.

### 2.6. Analyses

A total of 497 event attendees completed the survey; this is approximately 30 surveys per site across the 16 DHS and DPH health center sites. Surveys with missing data for several covariates were excluded from the analyses (n = 70). Some covariates had missing data that were retained because they were key demographic variables—e.g., a missing category for age (n = 51) and one for race/ethnicity (n = 20). Descriptive analyses were performed to describe the study outcomes, independent variables (in particular, frequency of attending free produce events at safety-net health center sites), and covariates. Multivariable regression analyses (models) were conducted to examine the associations between attending free produce events and the three outcomes of interest: fruit and vegetable consumption, household food insecurity, and SNAP participation. All analyses were carried out using SAS Version 9.4 (SAS Institute Inc., Cary, NC, USA). All study materials and instruments (i.e., survey questionnaire) were reviewed and approved by the DPH Institutional Review Board (IRB) prior to field administration (IRB #2014-09-535).

## 3. Results

Study attendees were predominately female (75%) and Hispanic or Latino/a/x (81%) (Table 1). Almost half were patients from the health center site facilitating the event. The attendees generally came from larger household sizes, with nearly 40% living in households of five or more people. Approximately 71% indicated they or someone in their household have been diagnosed with one or more chronic health conditions—heart disease, prediabetes, diabetes, hypertension (high blood pressure), and/or overweight/obese.

When they were asked how frequently they attended produce events at their health center site, 24% of the attendees reported that it was their first time, 54% came occasionally (less than one time per month), and 22% came frequently (about one time per month or more) (Table 1). On average, attendees reported consuming 2.5 servings of fruits and 2.5 servings of vegetables a day, with 55% of attendees consuming 5+ combined servings on an average day. Over 80% lived in food-insecure households. And only 23% reported that they were enrolled and actively participating in SNAP.

Table 2 shows attendees’ participation rates in public assistance programs and information about whether or not they were ever referred to apply and enroll in these programs. For those who self-identified as patients of the health center sites, 68% indicated they had received information and applied to Medi-Cal before, 28% had received information on SNAP, and 14% had received information on Supplemental Security Income (SSI); 72%, 26%, and 14% of this group reported being on Medicaid/Medi-Cal, SNAP, and SSI, respectively.

Table 3 shows the bivariate associations between the frequency of attending the free produce events and socio-demographic characteristics, fruit and vegetable servings consumed, household food insecurity, and SNAP participation. Compared to first-time attendees, those who attended the events occasionally or frequently consumed higher mean servings of fruit and vegetables on an average day. When fruit and vegetable servings were run as separate variables in the analysis, the same positive associations were identified for occasional or frequent attendees. The associations between the frequency of attending free produce events and household food insecurity and SNAP participation were not statistically significant. This bivariate analysis (Table 3) also showed that, relative to adults aged 35–64, older adults (age ≥ 65) consumed a higher number of fruit and vegetable servings. Among participants who reported they or someone in their household had at least one chronic health condition(s), 90% reported some level of household food insecurity.

Table 4 displays the multivariable regression analyses (models) for the three study outcomes. Those who attended the free produce events frequently consumed, on average, one additional serving of fruits and vegetables a day (*p* < 0.000) compared to first-time attendees, after adjusting for covariates. Compared to females, males consumed fewer servings of fruits and vegetables (*p* < 0.001). Those aged ≥65 consumed 0.64 (*p* < 0.003) more servings of fruits and vegetables and had higher odds of SNAP participation (Adjusted Odds Ratio 2.35, *p* < 0.004) than working-age adults (age group: 35–64).

## 4. Discussion

This study describes the demographic, health, and social characteristics of people who attended free produce events at safety-net health center sites in Los Angeles County, events facilitated by the single largest Medicaid-reliant health system in Southern California, USA (i.e., DHS). Nearly half of the attendees (48%) were patients of the health center sites that facilitated the events, while most (76%) reported being occasional (2–9 times) or frequent (≥10 times) visitors/beneficiaries of these food distributions. Most attendees who participated in the survey reported that they or someone in their household had at least one, if not more than one, chronic health condition(s), suggesting that this safety-net population has both high physical health and unmet social needs.

A high frequency of attending free produce events predicted (was found to be associated with) increased fruit and vegetable consumption among the attendees in this study—the coefficients or apparent strengths of the relationship, as expressed in the model, approximately doubled from the “occasional” to the “frequent” category when compared to the reference group, first-time attendees. While it is beyond the scope of this study to draw meaningful inferences, an intriguing hypothesis is that extended participation or exposure to this FAM strategy could lead to sustainable improvements in diet quality, literally by increasing most adults’ intake of fruits and vegetables by one additional cup a day. This finding is intriguing because it may have future implications and applications for changing health behaviors. A multi-site study of produce prescription programs, which subsidize fruits and vegetables purchased from grocery stores and farmers markets, found that fruit and vegetable intake similarly increased by almost a cup (0.85 cups) per day for adults with or at risk for cardiometabolic disease [13].

This study population reported high consumption of fruits and vegetables, with over half of the sample consuming 5+ servings in a day. This is higher than nationally representative data from federal and Los Angeles County data sources, which have shown a lower proportion of low-income adults (7–14%) consuming the recommended amount of fruits and vegetables a day [24,25]. The current study result is somewhat expected given that at the time of data collection the produce events had been implemented for several years, with some “regulars” lining up and waiting 1–2 h prior to the distribution of produce. The social interaction fostered by waiting in line and the routine of returning regularly may have contributed to increased access and improved consumption of fruits and vegetables.

An additional factor to consider, and inherent in the survey data collection, is social desirability bias, particularly when asking about dietary consumption [26,27]. While the study used a self-administered questionnaire, participants who self-reported personal health behaviors such as diet may have answered the survey question items with a biased viewpoint—i.e., predominantly favorable towards the program intervention because they were given free produce.

Offering free produce at safety-net health center sites has other key benefits as well. For instance, the strategy allows for more immediate consumer access to healthy food among safety-net populations who otherwise have high needs but low participation in public assistance programs. In this study, only about 23% of event attendees reported being enrolled and actively participating in SNAP, whereas nearly 84% indicated they were experiencing food insecurity. This mismatched ratio (disparity) between need (food insecurity) and remedy (public assistance usage) could help explain why a similar association between the frequency of attending free produce events at these sites and reduced household food insecurity or SNAP participation was not detected. Additionally, no association may have been identified for SNAP because while there was some promotion of SNAP at the produce events, it was limited and inconsistent across the sites. More research on these interacting factors is clearly needed in order to better understand and elucidate how these nuanced complexities influence the real-world implementation and health impacts of this FAM strategy [28,29].

## 5. Limitations

This study has several limitations. The county of Los Angeles is a vast, diverse region in the U.S., anchored by a complex healthcare landscape that comprises 7000+ licensed and certified healthcare facilities in the area [30,31,32]. From this diversity perspective, the sampled health center sites (n = 16) are a limited sample, representing only a non-generalizable cross-section of a larger, overall healthcare market in the region. The cross-sectional nature of this study prohibits any meaningful inferences about trends, patterns, or other longitudinal effects related to the free produce event strategy. Nevertheless, the in-person intercept survey did allow the DHS and DPH to capture previously unavailable information about attendees of these events.

Gaps in our understanding of the novelty and health impacts of onsite food distribution clearly warrant further investigation and evaluation [16,33,34]. Future research should consider the role and intensity of onsite produce events and how they affect healthcare utilization and cost. Qualitative research could similarly provide nuanced insights into key healthcare staff experiences that could be replicated and used to develop other innovations to promote healthy behaviors and improve health outcomes in the downstream.

## 6. Conclusions

Safety-net health centers are increasingly screening for food insecurity and providing patients with referrals to public assistance programs like SNAP. Unfortunately, not all individuals actively participate in or are eligible for these programs. Hence, onsite distributions of free produce at low-income health center sites are becoming a popular and promising complementary option for addressing this social need when traditional public assistance programming falls short. Health centers should participate in local food access coalitions and determine how best they can leverage partnerships with community organizations (e.g., food pantries, food banks, and others) to increase access to healthy food for their patient populations. A programmatic consideration may be to leverage other FAM interventions (e.g., produce prescription programs) in combination with onsite food resources (including produce pharmacies). Onsite distributions of free produce could and should be considered a viable FAM intervention for addressing hunger and food insecurity, especially under circumstances where patients are neither eligible nor using public assistance or when nutrition incentive/produce pharmacy programs are not readily available.

## Figures and Tables

**Table 1 nutrients-17-01849-t001:** Demographic and other characteristics of individuals who attended free produce events at safety-net health center sites in Los Angeles County, California, USA, 2020–2025 (n = 497).

Attendees	n	% or Mean (SD)
Gender		
Female	365	75.10
Male	121	24.90
Age		
18–34	36	7.24
35–64	255	51.31
≥65	155	31.19
Missing	51	10.26
Race/Ethnicity		
Black or African American	25	5.03
Hispanic or Latino/a/x	404	81.29
White	25	5.03
Other (Asian, AIAN, Multi-Race)	23	4.63
Missing	20	4.02
Patient at this Health Center Location		
No	244	52.36
Yes	222	47.64
Household Size		
1–2 people	110	23.50
3–4 people	182	38.89
5–6 people	143	30.56
7+ people	33	7.05
Diet-Related Chronic Health Condition in the Household		
No, does not have any listed chronic conditions	144	28.97
Yes, has chronic condition	353	71.03
High blood pressure	240	48.29
Heart disease	47	9.46
Diabetes, borderline diabetes, prediabetes	225	45.27
Overweight/obesity	106	21.33
Frequency of Attending Free Food (Produce) Events		
This is their first time	120	24.44
Occasional (2–9 times)	263	53.56
Frequent (≥10 times)	108	22.00
Outcome Variables		
Fruit Servings	497	2.57 (1.16)
Vegetable Servings	497	2.54 (1.21)
Fruit and Vegetable Servings	497	5.11 (2.17)
Household Food Insecurity ^A^		
No	81	16.30
Yes	416	83.70
Enrolled and Participating in the Supplemental Nutrition Assistance Program		
No	383	77.06
Yes	114	22.94

Note: AIAN = American Indian and Alaska Native, SD = standard deviation. ^A^ Assessed using the six-item U.S. Food Security Module—“food secure”: for those who scored 0–1 on the module; “food insecure”: for those who scored 2–6 on the module.

**Table 2 nutrients-17-01849-t002:** Participation in public assistance programs among patients of safety-net health center sites in Los Angeles County, California, USA, 2020–2025 (n = 222).

	Ever Received Referral from Healthcare Provider to Apply and Enroll to Program		Currently Participates	
	n	%	n	%
Supplemental Nutrition Assistance Program (SNAP)/CalFresh/Food Stamps	63	28.38	58	26.13
Medi-Cal	150	67.57	160	72.07
Special Supplemental Nutrition Program for Women, Infants and Children (WIC)	9	4.05	7	3.15
California Food Assistance Program	0	0.00	3	1.35
CalWorks (Temporary Assistance for Needy Families)	9	4.05	3	1.35
Housing Assistance Programs (Section 8)	0	0.00	11	4.95
Social Security Disability Insurance (SSDI)	12	5.41	10	4.50
Supplemental Security Income (SSI)	30	13.51	32	14.41
None of the above	50	22.52	28	12.61

**Table 3 nutrients-17-01849-t003:** Average fruit and vegetable servings consumed, household food insecurity, and SNAP participation by frequency of attending free produce events at safety-net health center sites in Los Angeles County, California, USA, 2020–2025.

	Fruit and Vegetable Servings		Household Food Insecurity			SNAP Participation		
	Mean (SD)	*p*-Value	No (%)	Yes (%)	*p*-Value	No (%)	Yes (%)	*p*-Value
Frequency of Attendance								
First time (ref)	4.45 (2.21)		14.17	85.83		75.00	25.00	
Occasional (2–9 times)	5.21 (2.06)	0.001	16.35	83.65	0.586	76.85	23.15	0.525
Frequent (≥10 times)	5.69 (2.25)	<0.000	15.74	84.26	0.739	77.95	22.05	0.774
Age								
35–64 (ref)	4.91 (2.09)		15.69	84.31		70.59	29.41	
18–34	5.08 (2.34)	0.671	25.00	75.00	0.218	75.00	25.00	0.882
≥65	5.49 (2.24)	0.008	13.55	86.45	0.384	83.87	16.13	0.016
Race/Ethnicity								
Hispanic or Latino/a/x (ref)	5.18 (2.19)		16.58	83.42		81.44	18.56	
Black or African American	4.70 (2.04)	0.283	20.00	80.00	0.658	48.00	52.00	0.000
White	4.34 (2.34)	0.061	8.00	92.00	0.270	68.00	32.00	0.105
Other (Asian, AIAN, Multi-Race)	5.09 (1.78)	0.933	17.39	82.61	0.959	65.22	34.78	0.001
Patient at this Clinic Location								
No (ref)	5.15 (2.16)		17.62	82.38		79.10	20.90	
Yes	5.05 (2.20)	0.606	14.86	85.14	0.421	73.87	26.13	0.184
Household Size								
1–2 people (ref)	5.00 (2.11)		17.27	82.73		69.09	30.91	
3–4 people	4.93 (2.14)	0.790	19.23	80.77	0.676	77.47	22.53	0.113
5–6 people	5.40 (2.19)	0.146	13.29	86.71	0.381	81.82	18.18	0.019
7+ people	5.33 (2.37)	0.432	15.15	84.85	0.775	72.73	27.27	0.690
Diet-Related Chronic Health Condition in Household								
No, does not have any listed chronic conditions (ref)	5.11 (2.04)		23.41	75.69		81.94	18.06	
Yes, has chronic condition	5.11 (2.23)	0.997	13.03	86.97	0.002	75.07	24.93	0.100

Note: AIAN = American Indian and Alaska Native, ref = reference category, SD = standard deviation, SNAP = Supplemental Nutrition Assistance Program.

**Table 4 nutrients-17-01849-t004:** Associations between frequency of attending free produce events at safety-net health center sites and attendees’ fruit and vegetable consumption, household food insecurity, and SNAP participation, Los Angeles County, California, USA, 2020–2025.

	Model 1 ^a^			Model 2 ^a^			Model 3 ^a^		
	Fruit and Vegetable Servings Consumed			Household Food Insecurity			SNAP Participation		
	Coef.	SE	*p*-Value	AOR	95% CI	*p*-Value	AOR	95% CI	*p*-Value
Frequency of attending free food events (ref = First time)									
Occasional (2–9 times)	0.53	0.22	0.016	1.52	[0.77, 3.01]	0.231	0.99	[0.56, 1.74]	0.960
Frequent (≥10 times)	1.00	0.26	0.000	0.64	[0.28, 1.43]	0.273	0.96	[0.49, 1.90]	0.908
Gender (ref = Female)									
Male	−0.69	0.21	0.001	0.69	[0.44, 1.36]	0.282	0.68	[0.40, 1.16]	0.159
Age (ref = 35–64)									
18–34	0.41	0.35	0.235	2.09	[0.87, 5.00]	0.099	1.07	[0.46, 2.53]	0.871
≥65	0.64	0.21	0.003	0.63	[0.32, 1.25]	0.186	2.35	[1.32, 4.18]	0.004
Missing	0.38	0.34	0.269	1.75	[0.73, 4.17]	0.211	2.97	[0.98, 9.01]	0.055
Race/Ethnicity (ref = Hispanic or Latino/a/x)									
Black	−0.42	0.42	0.314	1.68	[0.55, 5.14]	0.365	0.25	[0.10, 0.64]	0.004
White	−0.54	0.45	0.226	0.36	[0.05, 2.78]	0.325	0.77	[0.25, 2.35]	0.639
Other	0.08	0.41	0.848	0.88	[0.24, 3.18]	0.839	0.44	[0.17, 1.14]	0.092
Missing	−0.49	0.67	0.458	0.64	[0.06, 7.31]	0.721	0.55	[0.12, 2.47]	0.432
Patient at this Health Center Location (ref = No)									
Yes	−0.17	0.18	0.352	0.88	[0.51, 1.51]	0.643	0.69	[0.43, 1.11]	0.125
Enrolled and Participating in SNAP (ref = No)									
Yes	0.07	0.22	0.739	0.58	[0.29, 1.19]	0.140	N/A	N/A	N/A
Household Size (ref = 1–2 people)									
3–4 people	−0.24	0.24	0.324	1.02	[0.51, 2.07]	0.949	1.47	[0.80, 2.70]	0.215
5–6 people	0.21	0.26	0.429	0.60	[0.27, 1.34]	0.210	1.87	[0.95, 3.65]	0.068
7+ people	−0.01	0.38	0.983	0.91	[0.29, 2.83]	0.865	1.05	[0.41, 2.69]	0.914
n	427			427			427		

Note: AOR = Adjusted Odds Ratio, Coef. = coefficient, ref = reference category, SE = Standard Error, SNAP = Supplemental Nutrition Assistance Program. ^a^ Multivariable regression analyses, after adjusting for independent variables and covariates.

## Data Availability

The data presented in this article are not publicly accessible but are available upon request from and with approval from the corresponding author.

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
