# Peer review of "Fruit and Vegetable Consumption, Household Food Insecurity, and SNAP Participation Among Attendees of Free Produce Events at Safety-Net Health Center Sites"

_nutrients, 2025, doi:10.3390/nu17111849_

Round 1
Reviewer 1 Report
Comments and Suggestions for Authors
Thank you for the opportunity to review the article. LA area is well-know for high food prices. Free produce events can be an alternative that I have not seen covered in previous research. I have some concerns that I would recommend to cover:
- The results show that low-income people consume more than five portions a day of FV, while they experience very high food insecurity. This fact is, in some extent, opposite as previous literature. Of course, it maybe that free FV lead to such a great FV consumption increase. However, in general, low-income people that are not use to consume FV, they do not increase much consumption (even being free). It can also be the case the people overstate their actual FV consumption. I believe that the manuscript needs to include a discussion, and some analysis if the data allow it, on this.
- Also, previous literature has addressed the issue of support programs to low-income population (food stamps, vouchers, subsidies, etc...). The article needs to provide more context and discuss previous literature.
- The dataset is relative large (close to 500 observations). It would be interesting to conduct further analysis, for instance, it would be interesting to identify specific population segments (rather than have all population as a single sample).
I hope that my comments help to improve the article. Regards
Author Response
May 5, 2025
Dear Dr. Brown and Dr. Farmer,
We are pleased to resubmit our manuscript titled Fruit and vegetable consumption, household food insecurity, and SNAP participation among attendees of free produce events at safety-net health center sites for consideration as an Original Research paper in the Nutrients special issue: “Dietary Interventions to Advance Equity in Cardiometabolic Health.”
Below are our responses to the reviewers’ comments. Reviewer feedback has been invaluable. The suggested revisions have helped us greatly in improving the content and the writing of this paper. To better support the Background and Discussion sections, for example, we added additional peer-reviewed literature. To better describe the differences in study outcomes, we provided additional breakdown of the data by subgroup (age, race/ethnicity, see Table 3). And, in the Discussion section, we elaborated on results that are pertinent to the consumption of fruits and vegetables and the social desirability bias that may have influenced survey responses.
We believe that these edits and revisions, as suggested by the reviewers, have significantly strengthened our manuscript. Again, we thank you and the reviewers for this opportunity to revise and resubmit. We look forward to the next steps in this process.
Sincerely,
Alejandra Casillas MD, MSHS
Associate Professor of Medicine in Residence, Associate Vice Chair for Community Impact,
UCLA Department of Medicine- Division of General Internal Medicine and Health Services Research, Office of Community Engagement and Inclusive Excellence
UCLA David Geffen School of Medicine
---------------------------------------------------------------------------------------------------------------------------------
Italicized are the reviewer’s comments. Authors’ response(s) are in plain text.
REVIEWERS’ COMMENTS
Reviewer 1: Thank you for the opportunity to review the article. LA area is well-known for high food prices. Free produce events can be an alternative that I have not seen covered in previous research. I have some concerns that I would recommend to cover:
1.1 The results show that low-income people consume more than five portions a day of FV, while they experience very high food insecurity. This fact is, in some extent, opposite as previous literature. Of course, it maybe that free FV lead to such a great FV consumption increase. However, in general, low-income people that are not use to consume FV, they do not increase much consumption (even being free). It can also be the case the people overstate their actual FV consumption. I believe that the manuscript needs to include a discussion, and some analysis if the data allow it, on this.
We thank the reviewer for raising this point. We recognize that the complexities around assessing diet and the potential for social desirability bias are some of the factors that may have swayed respondents to overstate their fruit and vegetable (FV) consumption, especially when they were likely caught up in the moment—happy to have received fruits and vegetables for free. In asking about FV consumption in the survey, we did provide standard definitions about serving size—e.g., “a serving as approximately the size of fist”—to help reduce ambiguity in the reporting. As such, volume/serving size would not and should not have been a major contributor to this bias. When the responses to the FV consumption question were collapsed into a binary variable we did find that 55% of the study sample reported an intake of 5+ fruits and vegetables/day; this was high and worth noting in the paper. We added a new paragraph to further discuss this in the Discussion. See line(s) 281-296.
1.2 Also, previous literature has addressed the issue of support programs to low-income population (food stamps, vouchers, subsidies, etc...). The article needs to provide more context and discuss previous literature.
We thank the reviewer for this suggestion. In response, we added additional literature about public assistance programs in the Introduction. See line(s) 63: Challenges with low patient participation in traditional public assistance programs, such as SNAP, remain despite emerging research showing the positive impacts of SNAP on health outcomes, food security, and healthcare utilization [8,9].
1.3 The dataset is relative large (close to 500 observations). It would be interesting to conduct further analysis, for instance, it would be interesting to identify specific population segments (rather than have all population as a single sample).
We thank the reviewer for this suggestion. In response, we added additional breakdown results by subgroup (population segments) in Table 3.
Reviewer 2: Brief Summary This study examines free agricultural product distribution events at healthcare centers in Los Angeles County and their association with fruit and vegetable consumption, food insecurity and participation in the SNAP program. The results show that frequent participants in the events consumed on average one more serving of fruit and vegetables per day compared to new participants, while no association was found with food insecurity or SNAP participation.
We thank the reviewer for these comments. See more detailed responses below.
2.1 General Comments The manuscript addresses a relevant topic, providing empirical data on a "Food as Medicine" intervention that has been little studied. However, the introduction could include more details on previous studies on similar interventions. It is interesting, but appears unclear, I try to explain what I would recommend to review better:
We agree with the reviewer that the Introduction could use further details and context. In response, we added a definition for the “Food as Medicine” concept and described the range of strategies or programs that are typically considered FAM interventions. We also included new literature on the positive implications of this research (our present work included among them) on patient health and social outcomes.
Specific Comments
2.2 Lines 47-48: It is stated that "About half of all adult Americans have one or more preventable chronic diseases—many of them are attributed to poor diet". This statement should be supported with more specific and updated data. What is the exact prevalence of preventable diet-related chronic diseases?
We thank the reviewer for this comment. We added more specific information about the prevalence of cardiovascular disease and diabetes. See Line 47: Approximately 40% of households in the United States (U.S.) are food insecure, with about 40% of the U.S. population having some form of cardiovascular disease and 50% with prediabetes or diabetes much of which can be attributed to poor diet [1-4].
2.3 Lines 61-62: The term "Food as Medicine" (FAM) is introduced without a clear definition. It would be useful to provide a brief definition of this concept.
We thank the reviewer for this suggestion. On Line 66, we added a definition for the “Food as Medicine” concept.
2.4 Lines 134-139: Social desirability bias in measuring fruit and vegetable consumption should be discussed in the Limitations section, which is not extensive and would deserve more attention to areas that are critical in the study.
We thank the reviewer for this suggestion. Because the other reviewer also raised this social desirability bias issue, we added additional text in the Discussion section (rather than in the Limitations section) to discuss the potential influences of social desirability bias on study respondents’ reporting of fruit and vegetable consumption. See line(s) 291 - 296.
2.5 Table 3: The p-values should be formatted consistently. Sometimes in fact you insert one decimal digit and sometimes two, especially when the last digit is 0, sometimes you indicate it and sometimes not...
We thank the reviewer for identifying these discrepancies. We corrected the formatting.
2.6 Lines 216-222: The discussion of non-significant results is too brief. What hypotheses can be formulated to explain these results?
We thank the reviewer for this comment. In the Discussion section we provided a plausible explanation for the non-significant finding related to SNAP. See line(s): 305 Additionally, no association may have been identified for SNAP because while there was some promotion of SNAP at the produce events, it was limited and inconsistent across the sites.
2.7 Lines 241-250: A more explicit comparison with similar interventions would be useful.
We thank the reviewer for this suggestion. In the Discussion section, starting Line 277, we included more details about how our results align with those of other FAM intervention/produce prescription work, which suggest that these types of interventions can help regularly increase access to produce.
2.8 Lines 289-298: Add more specific recommendations for practice and future research.
In response to the reviewer’s suggestion, we added several recommendations about future research in the Limitations section, and several policy/practice recommendations in the Conclusions section.
2.9 Furthermore, it is not specified whether this questionnaire was previously validated or if it was developed specifically for this study. This aspect should be clarified because I did not understand it.
While most of the question items in the questionnaire (study instrument) were standard questions that are often used in surveys, some of the question items were developed specifically for this program evaluation. For example, due to an interest by the program, two questions were asked about fruit and vegetables with a recall timeframe of ‘in the past 24 hours’. See line 145 for more details. Where feasible, adaptations and/or use of the various question items were carried out with administration of the survey employing an intercept-style survey approach—i.e., needed to be quick and easy to understand).
2.10 Finally I noticed a certain repetitiveness in the conclusions that are almost verbatim from the abstract. I would suggest elaborating more on the conclusions section to provide a more reflective synthesis of the results and their implications.
We thank the reviewer for this comment. In response, we added more details in the Conclusions section.
I look forward to reading an improved version. Thank you.
Reviewer 2 Report
Comments and Suggestions for Authors
Dear corresponding Author, thank you for submitting your work to Nutrients journal and congratulations on your research.
Brief Summary This study examines free agricultural product distribution events at healthcare centers in Los Angeles County and their association with fruit and vegetable consumption, food insecurity and participation in the SNAP program. The results show that frequent participants in the events consumed on average one more serving of fruit and vegetables per day compared to new participants, while no association was found with food insecurity or SNAP participation.
General Comments The manuscript addresses a relevant topic, providing empirical data on a "Food as Medicine" intervention that has been little studied. However, the introduction could include more details on previous studies on similar interventions. It is interesting, but appears unclear, I try to explain what I would recommend to review better:
Specific Comments
- Lines 47-48: It is stated that "About half of all adult Americans have one or more preventable chronic diseases—many of them are attributed to poor diet". This statement should be supported with more specific and updated data. What is the exact prevalence of preventable diet-related chronic diseases?
- Lines 61-62: The term "Food as Medicine" (FAM) is introduced without a clear definition. It would be useful to provide a brief definition of this concept.
- Lines 134-139: Social desirability bias in measuring fruit and vegetable consumption should be discussed in the Limitations section, which is not extensive and would deserve more attention to areas that are critical in the study.
- Table 3: The p-values should be formatted consistently. Sometimes in fact you insert one decimal digit and sometimes two, especially when the last digit is 0, sometimes you indicate it and sometimes not...
- Lines 216-222: The discussion of non-significant results is too brief. What hypotheses can be formulated to explain these results?
- Lines 241-250: A more explicit comparison with similar interventions would be useful.
- Lines 289-298: Add more specific recommendations for practice and future research. Furtheremore, it is not specified whether this questionnaire was previously validated or if it was developed specifically for this study. This aspect should be clarified because I did not understand it.
Finally I noticed a certain repetitiveness in the conclusions that are almost verbatim from the abstract. I would suggest elaborating more on the conclusions section to provide a more reflective synthesis of the results and their implications.
I look forward to reading an improved version.
Author Response

(The authors gave the same response as above.)

Round 2
Reviewer 1 Report
Comments and Suggestions for Authors
Most of my previous concerns were addressed in the revised manuscript. I still find that the review of the US evidence is too short. Overall, it is ok. Regards
Author Response
Reviewer 1: Most of my previous concerns were addressed in the revised manuscript. I still find that the review of the US evidence is too short. Overall, it is ok.
We thank the reviewer for reviewing our revised manuscript and for providing this additional comment about the need for additional discussion about the evidence. In the Introduction section we added references related to onsite food distributions and other FAM strategies, including studies that discussed the future directions of this work. Starting on Line 77 we added: “Patients, for instance, who visited a hospital-based food pantry, have expressed comfort, convenience, and less stigma…food pantries.” Starting on Line 84 we added: “Despite FAM leaders and practitioners calling for greater integration of these programs into health-related settings, these alternative intervention options…”
Additionally, in the Limitations section, we discussed similar onsite food distribution and related strategies from the literature, and indicated the need for more research in this area—please see paragraph starting on Line 338: “Gaps in our understanding of the novelty and health impacts of onsite food distribution clearly warrant further investigation and evaluation. Future research...”
Reviewer 2 Report
Comments and Suggestions for Authors
"I have carefully read the changes made by the authors and believe that the effort to improve the work has been worthwhile. I therefore consider it suitable for acceptance in its current form
Author Response
Reviewer 2: I have carefully read the changes made by the authors and believe that the effort to improve the work has been worthwhile. I therefore consider it suitable for acceptance in its current form
We thank the reviewer for their time in reviewing our paper and for their helpful comments.